# Ship Identification and Characterization in Sentinel-1 SAR Images with Multi-Task Deep Learning

**Clément Dechesne [1], Sébastien Lefèvre [2,*], Rodolphe Vadaine [3], Guillaume Hajduch [3] and Ronan Fablet [1]**

1   IMT Atlantique—Lab-STICC, UMR CNRS 6285, 29238 Brest, France; clement.dechesne@cnes.fr (C.D.); ronan.fablet@imt-atlantique.fr (R.F.)
2   Univ. Bretagne Sud—IRISA, UMR CNRS 6074, 56017 Vannes, France
3   Collecte Localisation Satellites, 29280 Brest, France; rvadaine@groupcls.com (R.V.); ghajduch@groupcls.com (G.H.)
*   Correspondence: sebastien.lefevre@irisa.fr

**Abstract:** The monitoring and surveillance of maritime activities are critical issues in both military and civilian fields, including among others fisheries' monitoring, maritime traffic surveillance, coastal and at-sea safety operations, and tactical situations. In operational contexts, ship detection and identification is traditionally performed by a human observer who identifies all kinds of ships from a visual analysis of remotely sensed images. Such a task is very time consuming and cannot be conducted at a very large scale, while Sentinel-1 SAR data now provide a regular and worldwide coverage. Meanwhile, with the emergence of GPUs, deep learning methods are now established as state-of-the-art solutions for computer vision, replacing human intervention in many contexts. They have been shown to be adapted for ship detection, most often with very high resolution SAR or optical imagery. In this paper, we go one step further and investigate a deep neural network for the joint classification and characterization of ships from SAR Sentinel-1 data. We benefit from the synergies between AIS (Automatic Identification System) and Sentinel-1 data to build significant training datasets. We design a multi-task neural network architecture composed of one joint convolutional network connected to three task specific networks, namely for ship detection, classification, and length estimation. The experimental assessment shows that our network provides promising results, with accurate classification and length performance (classification overall accuracy: 97.25%, mean length error: 4.65 m ± 8.55 m).

**Keywords:** deep neural network; Sentinel-1 SAR images; ship identification; ship characterization; multi-task learning

## 1. Introduction

Deep learning is considered as one of the major breakthroughs related to big data and computer vision [1]. It has become very popular and successful in many fields including remote sensing [2]. Deep learning is a paradigm for representation learning and is based on multiple levels of information. When applied on visual data such as images, it is usually achieved by means of convolutional neural networks. These networks consist of multiple layers (such as convolution, pooling, fully connected, and normalization layers) aiming to transform original data (raw input) into higher level semantic representation. With the composition of enough such elementary operations, very complex functions can be learned. For classification tasks, higher level representation layers amplify aspects of the input that are important for discrimination and discard irrelevant variations. For humans, it is simple through visual inspection to know what objects are in an image, where they are, and how they interact

in a very fast and accurate way, allowing performing complex tasks. Fast and accurate algorithms for object detection are thus sought to allow computers to perform such tasks, at a much larger scale than humans can achieve.

Ship detection and classification have been extensively addressed with traditional pattern recognition techniques for optical images. Zhu et al. [3] and Antelo et al. [4] extracted handcrafted features from images such as shapes, textures, and physical properties, while Chen et al. [5] and Wang et al. [6] exploited dynamic Bayesian networks to classify different kinds of ships. Such extracted features are known for their lack of robustness, which can raise challenges in practical applications (e.g., they may lead to poor performances when the images are corrupted by blur, distortion, or illumination, which are common artifacts in remote sensing). Furthermore, they cannot overcome the issues raised by big data such as image variabilities (i.e., ships of the same type may have different shapes, colors, sizes, etc.) and data volume. Recently, following the emergence of deep learning, an autoencoder based deep neural network combined with extreme learning machine was proposed [7] and outperformed some other methods using SPOT-5 spaceborne optical images for ship detection.

Compared with optical remote sensing, satellite SAR imaging appears more suited for maritime traffic surveillance in operational contexts, as it is not critically affected by weather conditions and day-night cycles. In this context, open-source Sentinel-1 SAR data are particularly appealing. Almost all coastal zones and shipping routes are covered by Interferometric Wide Swath Mode (IW), while the Extra-Wide Swath Mode (EW) acquires data over open oceans, providing a global coverage for sea oriented applications. Such images, combined with the Automatic Identification System (AIS), represent a large amount of data that can be employed for training deep learning models [8]. AIS provides meaningful and relevant information about ships (such as position, type, length, rate of turn, speed over ground, etc.). The combination of these two data sources could leverage new applications to the detection and estimation of ship parameters from SAR images, which remains a very challenging task. Indeed, detecting inshore and offshore ships is critical in both military and civilian fields (e.g., for monitoring of fisheries, management of maritime traffic, safety of coast and sea, etc.). In operational contexts, the approaches used so far still rely on manual visual interpretations that are time consuming, possibly error prone, and definitely irrelevant to scale up to the available data streams. On the contrary, the availability of satellite data such as Sentinel-1 SAR makes possible the exploration of efficient and accurate learning based schemes.

One may however consider with care AIS data as they involve specific features. AIS is mandatory for large vessels (e.g., >500 GT, passenger vessels). As such, it provides representative vessel datasets for international maritime traffic, but may not cover some maritime activities (e.g., small fishing vessels). Though not authorized, ships can easily turn off their AIS and/or spoof their identity. While AIS tracking strategies [9] may be considered to address missing track segments, the evaluation of spoofing behavior is a complex task. Iphar et al. [10] evaluated that amongst ships with AIS, about 6% have no specified type, and 3% are only described as "vessels". Besides, respectively 47% and 18% of the vessels may involve uncertain length and beam data. These points should be considered with care in the analysis of AIS datasets, especially when considering learning strategies as addressed in this work.

Among existing methods for ship detection in SAR images, Constant False Alarm Rate (CFAR) based methods have been widely used [11,12]. The advantage of such methods is their reliability and high efficiency. Using AIS information along with SAR images significantly improves ship detection performance [13]. As the choice of features has an impact on the performance of discrimination, deep neural networks have recently taken the lead thanks to their ability to extract (or learn) features that are richer than handcrafted (or expert) features. In [14], a framework named Sea-Land Segmentation based Convolutional Neural Network (SLS-CNN) was proposed for ship detection, combined with the use of saliency computation. A modified Faster R-CNN based on the CFAR algorithm for SAR ship detection was proposed in [15] with good detection performance. In [16], Texture Features extracted from SAR images were fed into Artificial Neural Networks (TF-ANN) to discriminate ship pixels from sea ones.

Schwegmann et al. [17] employed a highway network for ship detection in SAR images and achieved good results, especially in reducing the false detection rate. These state-of-the-art approaches focused on ship detection in SAR images. In this paper, we aim to go beyond ship detection and investigate higher level tasks, namely the identification of ship types (also known as classification) and their length estimation, which to our knowledge remain poorly addressed using learning based frameworks.

The problem of ship length estimation from SAR images was briefly discussed in [18,19]. In [18], the best shape of a ship was extracted from a SAR image using inertia tensors. The estimated shape allowed obtaining the ship length. However, the absence of a ground truth does not allow validating the accuracy of this method. In [19], a three step method was proposed in order to extract a rectangle that would be the reference model for ship length estimation. The method produced good results (mean absolute error: $30\,\text{m} \pm 36.6\,\text{m}$). However, the results were presented on a limited dataset (only 127 ships), and their generalization may be questioned.

In this paper, we propose a method based on deep learning for ship identification and characterization with the synergetic use of Sentinel-1 SAR images and AIS data.

## 2. Material and Methods

The proposed framework combined the creation of a reference ground truthed dataset using AIS-SAR synergies and the design of a multi-task deep learning model. In this section, we first introduce the proposed multi-task neural network architecture, which jointly addresses ship detection, classification, and length estimation. Second, we describe the training framework in terms of the considered training losses and of the implemented optimization scheme. Third, we detail the creation of the considered reference datasets, including how we tackled data augmentation and class imbalance issues, which have been shown to be critical for the learning process.

### 2.1. Proposed Framework

The proposed multi-task framework was based on two stages, with a first common part and then three task oriented branches for ship detection, classification, and length estimation, respectively (see Figure 1). The first part was a convolutional network made of 5 layers. It was followed by the task oriented branches. All these branches were made of convolutional layers followed by fully connected layers (the number of which depended on the complexity of the task). For the detection task, the output consisted of a pixel-wise probability map of the presence of ships. It only required 1 fully connected layer after the 4 convolutional layers. For the classification task, we considered 4 or 5 ship classes (cargo, tanker, fishing, passenger, and optionally tug). The branch also required 4 convolutional layers and 2 fully connected layers. The last task was related to the length estimation. This branch was composed of 4 convolutional layers and 5 fully connected layers.

This architecture was inspired by state-of-the-art architectures [20–22]. The number of layers was chosen to be similar to the first layers of the VGG network [22]. All the activations of the convolutional layers and fully connected layers were ReLU [23]. Other activation functions were employed for the output layers: a sigmoid for the detection, a softmax activation for the classification, and a linear activation employed for the length estimation; further details are presented in Section 2.2. We may emphasize that our model included a detection component. Though this was not a targeted operational objective in our context, it was shown to improve the performance for the other tasks (see Section 3.3).

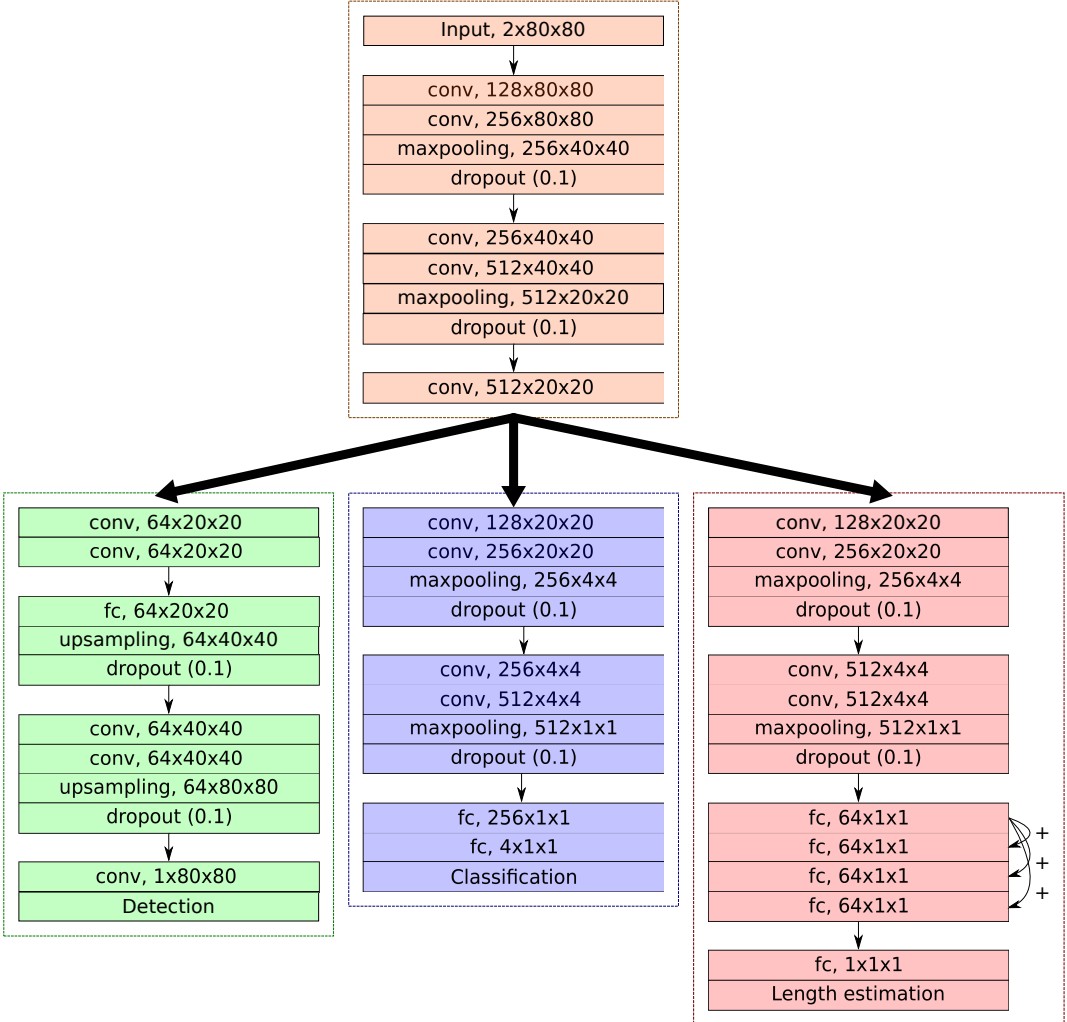

**Figure 1.** Proposed multi-task architecture for ship detection, classification (4 classes) and length estimation from a Sentinel-1 SAR image.

### 2.2. Training Procedure

We describe below the considered training strategy, especially the training losses considered for each task specific component. The proposed end-to-end learning scheme combined task specific training losses as follows:

- Detection loss: The detection output was the ship presence probability. We employed a binary cross-entropy loss, which is defined by:

$$L_{det} = -\frac{1}{N} \sum_{n=1}^{N} \sum_{k \in I} (y_k \log(p(k)) + (1 - y_k) \log(1 - p(k)), \tag{1}$$

where $N$ is the number of samples, $k$ is a pixel of the output detection image $I$, $y_k$ is the ground truth of ship presence (0 or 1), and $p(k)$ is the predicted probability of ship presence. It is a usual loss function for binary classification tasks [24].

- Classification loss: The output for the last classification layer was the probability that the input image corresponded to one of the considered ship types. We used here the categorical cross-entropy loss:

$$L_{class} = -\frac{1}{N} \sum_{n=1}^{N} \sum_{c=1}^{n_c} (y_{o,c} \log(p_{o,c})), \tag{2}$$

where $N$ is the number of samples, $n_c$ is the number of classes (here, $n_c = 4$ or $n_c = 5$), $y_{o,c}$ is a binary indicator (0 or 1) if class label $c$ is the correct classification for observation $o$, and $p_{o,c}$ is the predicted probability for the observation $o$ to belong to class $c$. It is a widely used loss function for multiclass classification tasks [25,26].

- Length estimation loss: In the length estimation network, the 4 fully connected layers of shape ($64 \times 1 \times 1$) were connected to each other (see Figure 1). The idea was to propagate the difference between the first layer and the current layer and was related to residual learning [27]. We used here the mean squared error, defined as:

$$L_{length} = \frac{1}{N} \sum_{n=1}^{N} (l_{pred} - l_{true})^2,$$ 

(3)

where $N$ is the number of samples, $l_{pred}$ is the predicted length, and $l_{true}$ is the true length.

Overall, we define the loss function of the whole network as:

$$L = L_{det} + L_{class} + L_{length}.$$

(4)

Each specific loss employed to design the loss of the whole network could have been weighted. Nevertheless, we observed no significant effect of such a weighting scheme. Thus, we decided to rely on a simple combination through adding the different task dedicated losses, giving the same importance to each task. Our network was trained end-to-end using the RMSProp optimizer [28]. The weights of the network were updated by using a learning rate of $1 \times 10^{-4}$ and a learning rate decay over each update of $1 \times 10^{-6}$ over the 500 iterations. Such parameterization showed good results for our characterization tasks.

### 2.3. Creation of Reference Datasets

With a view toward implementing deep learning strategies, we first address the creation of reference datasets from the synergy between AIS data and Sentinel-1 SAR data. AIS transceiver sent data every 2 to 10 s. These data mainly consisted of the positional accuracy (up to 0.0001 min precision) and the course over ground (relative to True North to 0.1°). For a given SAR image, one could interpolate AIS data from the associated acquisition time. Thus, it was possible to know the precise location of the ships in the SAR image and the related information (in our case, length and type). The footprint of the ship was obtained by thresholding the SAR image in the area where it was located (the brightest pixel of the image). Since the database was very unbalanced in terms of class distribution, a strategy was also proposed in order to enlarge the training set with translations and rotations, which is a standard procedure for database enlargement (also known as data augmentation). Concurrently to our work, a similar database was proposed in [29]. We also evaluated our framework with this dataset (see Section 3.5).

In our experiments, we considered a dataset composed of 18,894 raw SAR images of size $400 \times 400$ pixels with a 10 m resolution. The polarization of the images were either HH (proportion of Horizontally transmitted waves that return Horizontally) or VV (proportion of Vertically transmitted waves that return Vertically). Polarization had a significant effect on SAR backscatter. However, our goal was to allow us to process any Sentinel-1 SAR images. We thus considered any HH and VV polarized image without prior information on the type of polarization. Each image was accompanied with the incidence angle since it impacted the backscatter intensity of the signal. For the proposed architecture, the input was a 2 band image (backscatter intensity and incidence angle). Thus, we did not use any pre-trained network since we assumed that they could not handle such input data. We relied on the Automatic Identification System (AIS) to extract images that contained a ship in their center. AIS also provided us with information about the ship type and length. As stated before, AIS may have been corrupted (e.g., with spoofing), so when creating the database, we only considered ships that responded to the two following criteria: (i) their type was clearly defined (i.e., they belonged to the

retained classes); (ii) their length was greater than 0 and smaller than 400 m (the largest ship in the world). Besides, the SAR images we selected were acquired over European waters, where we expected AIS data to be of higher quality compared with other maritime areas.

The dataset was strongly imbalanced, amongst the 5 classes (tanker, cargo, fishing, passenger and tug), cargo was the most represented (10,196 instances), while tug was the least represented (only 444 instances). The class distribution is detailed in Figure 2 and Table 1. The length distribution showed that tanker, cargo, and passenger ships had similar length distributions. fishing ships had relatively small lengths, while tug ships' lengths were intermediate.

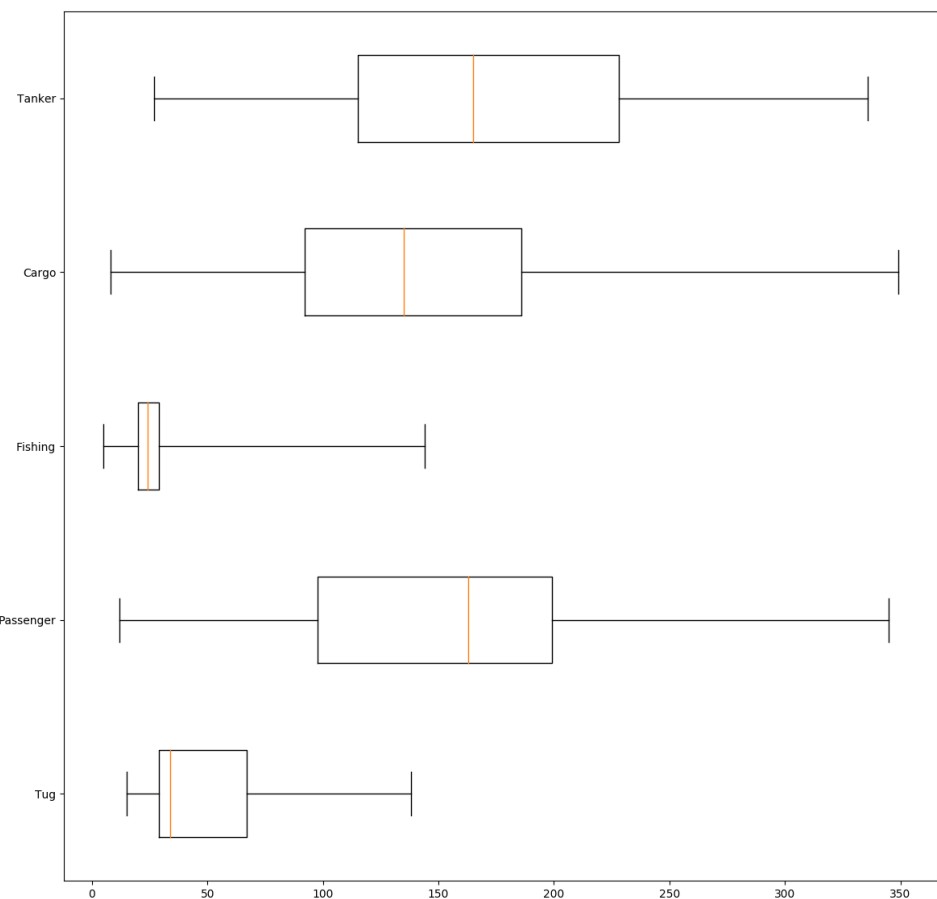

**Figure 2.** Boxplot of the length distribution for each class in our dataset. Length data are given in meters.

**Table 1.** Length distribution and number of samples for each class in our dataset.

|  | Tanker | Cargo | Fishing | Passenger | Tug |
|---|---|---|---|---|---|
| Number of samples | 4737 | 10196 | 2664 | 1071 | 444 |
| Length mean (m) | 168.4 | 146.5 | 26.6 | 153.5 | 47.3 |
| Length standard deviation (m) | 64.6 | 60.8 | 12.2 | 68.9 | 26.0 |
| Number of augmented samples (4 classes) | 263 | 0 | 2336 | 3929 | - |
| Number of augmented samples (5 classes) | 0 | 0 | 1336 | 2929 | 3556 |

To account for class imbalance [30], we applied data augmentation with translations and rotations. We first performed a rotation of a random angle centered on the brightest pixel of the SAR image (the center of the ship) and then performed a random translation. The same transformation was applied to the incidence angle image. The images employed to train the networks were of size $80 \times 80$ pixels. They contained ships (not necessarily in their center; see Figure 3). The ship footprint ground truth was generated by thresholding the SAR image since we precisely knew the location of the ship (i.e., it was the brightest pixel of the SAR image; see Figure 4). The obtained footprint was not perfect (see Figure 4b), but was shown to be sufficient to train the network. Let us note that a CFAR approach could have been employed in order to extract more precisely the ship footprint [11]. However, since our goal was not to detect ships, a coarse ship footprint was sufficient. We considered 2 configurations for the databases; a 4 class database, employed to compare our baseline to other state-of-the-art approaches (namely MLP and R-CNN), and a 5 class database in order to evaluate how our network responded with more classes. Each database was composed of 20,000 images of $80 \times 80$ pixels, with the same amount of samples per class (5000 per class for the 4 class database and 4000 per class for the 5 class database). The networks were trained with 16,000 images, and the remaining 4000 were used for validation. Throughout the data augmentation process, we ensured that images could be seen either in the training or validation set, but not in both. Ships with no AIS signal were not considered in our dataset (neither to train nor evaluate our model), since our strategy to build the dataset relied on matching the AIS signal with SAR imagery. However, once a model was trained, it could perform in an operational settings to detect ships with no AIS (this is indeed one of our long term goals).

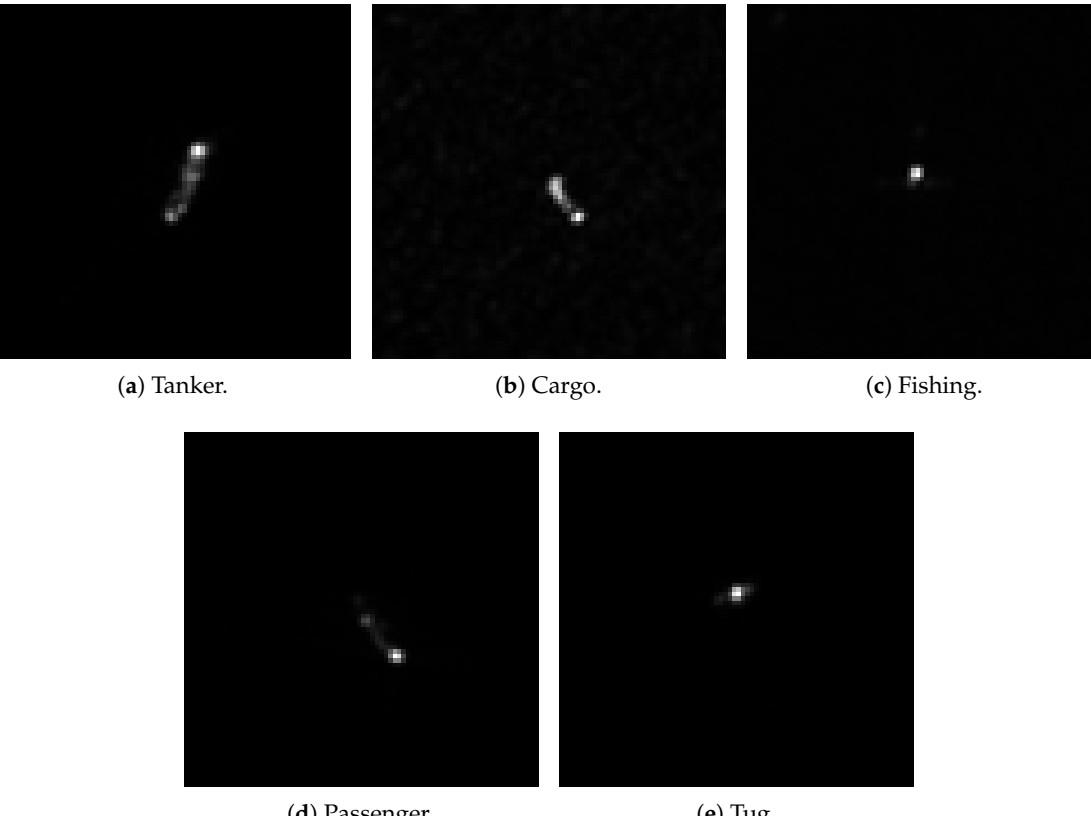

(**a**) Tanker.  (**b**) Cargo.  (**c**) Fishing.

(**d**) Passenger.  (**e**) Tug.

**Figure 3.** Examples of SAR image (with backscatter intensity) for each ship type of the collected database.

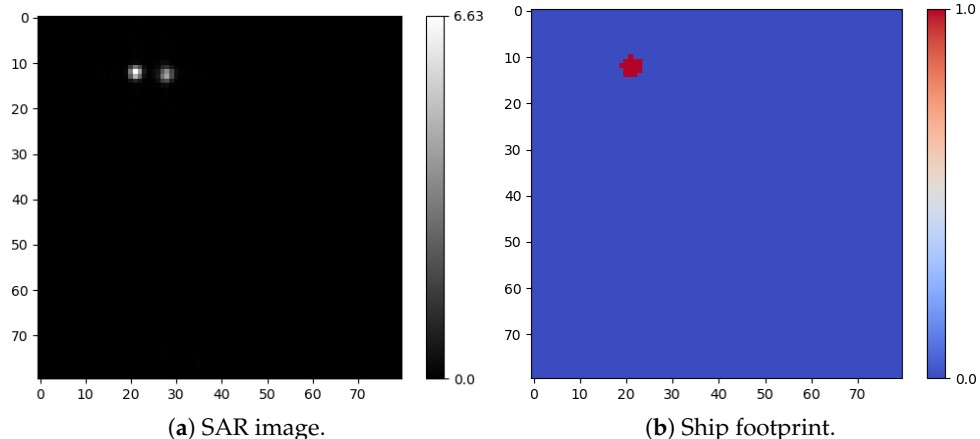

(**a**) SAR image.      (**b**) Ship footprint.

**Figure 4.** Example of a SAR image (with backscatter intensity) and associated ship footprint (best viewed in color).

## 3. Results

We ran all numerical experiments on a PC with an NVidia GTX 1080 Ti, an Intel Xeon W-2145 CPU 3.70 GHz, and 64 GB RAM (with a Keras [31] implementation). We evaluated the proposed framework with respect to other popular deep learning based solutions. We first considered a Multi-Layer Perceptron (MLP) [32] with only one hidden layer with 128 hidden units. The MLP is the most simple network that can be proposed for the desired task and could be a good basis in order to evaluate the performance of our network. We also designed a R-CNN (Regions with CNN features) [33] network in order to extract ship bounding boxes along with classification. Even if the R-CNN based bounding boxes did not allow precisely measuring the ship length, they could provide a good basis for its estimation. R-CNN is a state-of-the-art algorithm for object detection and classification [33]. Thus, it is worth being compared with our proposed model. The R-CNN had a very simple architecture, as presented in Figure 5. The networks were trained using 16,000 images from the augmented dataset, and the remaining 4000 images were used for validation.

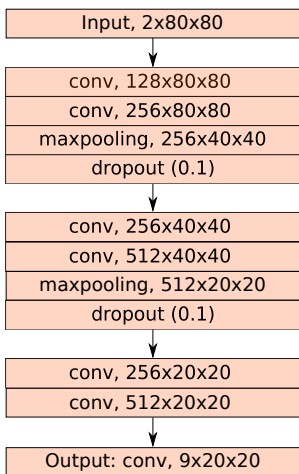

**Figure 5.** R-CNN architecture considered for ship classification.

The evaluation of the models was performed using several metrics. The classification task was assessed through the confusion matrix, giving, for each class and overall, several metrics. The Intersection over Union (IoU or Jaccard index) [34] measures similarity between finite sample sets and is defined as the size of the intersection divided by the size of the union of the sample sets. It was designed for the evaluation of object detection. The F-score is the harmonic mean of precision and recall, and it reaches its

best value at one and the worst at zero. The Kappa coefficient [35] ($\kappa$) is generated from a statistical test to evaluate the accuracy of a classification. Kappa essentially evaluates how well the classification performs as compared to just randomly assigning values (i.e., did the classification do better than randomness?). The Kappa coefficient can range from $-1$ to one. A value of zero (respectively $-1$ or one) indicates that the classification is no better (respectively worse or better) than a random classification. For the length estimation task, the mean error (and its standard deviation) were employed. For a ship $k$, the length error is defined as $e_k = l_{k,pred} - l_{k,true}$, where $l_{k,pred}$ is the predicted length and $l_{k,true}$ is the actual length. The mean error $m_{err\_length}$ (respectively the standard deviation $stdev_{err\_length}$) is the mean (respectively the standard deviation) of all the $e_k$. We further refer to the mean error as $m_{err\_length} \pm stdev_{err\_length}$.

### 3.1. MLP Model

For an $80 \times 80$ image, the MLP ran at 2000 frames per second. The whole training took about one hour. The testing took less than a minute. It produced very poor results. Indeed, the overall accuracy for classification was 25%, which means that the classifier assigned the same class to all the images (see Table 2). The length estimation was also rather inaccurate, the ship length being underestimated with a very large standard deviation (mean error: $-7.5\,\text{m} \pm 128\,\text{m}$).

**Table 2.** Confusion matrix and accuracy metrics for the MLP with 4 classes.

| | Confusion Matrix | | | | |
|---|---|---|---|---|---|
| **Ground Truth** — Prediction | **Tanker** | **Cargo** | **Fishing** | **Passenger** | **Precision** |
| Tanker | 1000 | 0 | 0 | 0 | 100.0 |
| Cargo | 1000 | 0 | 0 | 0 | 0.0 |
| Fishing | 1000 | 0 | 0 | 0 | 0 |
| Passenger | 1000 | 0 | 0 | 0 | 0.0 |
| Recall | 25.0 | - | - | - | |
| | Accuracy metrics | | | | |
| Label | Tanker | Cargo | Fishing | Passenger | Overall |
| IoU | 25.0 | 0.0 | 0.0 | 0.0 | 6.25 |
| F-Score | 40.0 | - | - | - | 10.00 |
| Accuracy | 25.0 | 75.0 | 75.0 | 75.0 | 25.00 |
| $\kappa$ | 0.0 | 0.0 | 0.0 | 0.0 | 0.25 |

### 3.2. R-CNN Model

For an $80 \times 80$ image, the R-CNN ran at 333 frames per second. The whole training took about 6.5 h and the testing about a minute. It produced better results than the MLP. The network estimated the 4 corners of the bounding box. As the ground truth for bounding boxes was obtained from the ship footprint extracted by thresholding the SAR image, it might not be well defined (see Figure 6c,d). In Figure 6c, the bounding box is well centered on the ship, but has the wrong size. In Figure 6d, the bounding box is also not well sized and accounts for the brightest part of the ship. We recall that the detection task was not our main objective, but rather regarded as a means to better constrain the training of the models. The R-CNN had a classification overall accuracy of 89.29%. Several other metrics are presented in Table 3.

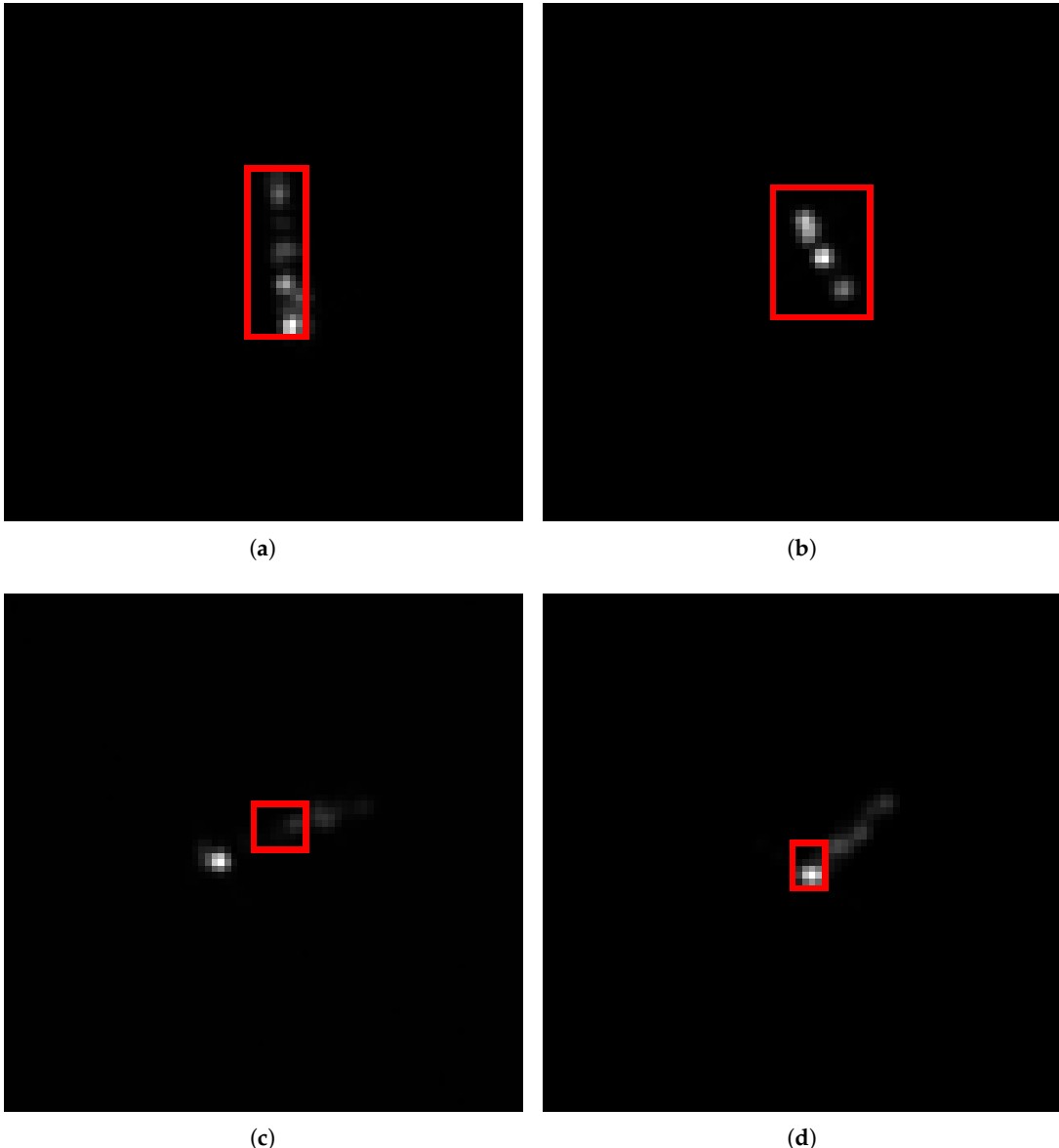

**Figure 6.** Illustration of the detection and classification performance of the evaluated R-CNN model: each subpanel depicts a SAR image with the superimposed detected bounding box. (**a**) Accurate bounding box superimposed on the SAR image. The ship type (passenger) is well predicted. (**b**) Accurate bounding box superimposed on the SAR image. The ship type (tanker) is well predicted. (**c**) Inaccurate bounding box superimposed on the SAR image. The ship type predicted (fishing) is not the right one (cargo). (**d**) Inaccurate bounding box superimposed on the SAR image. The ship type (tanker) is well predicted.

### 3.3. Our Network

For an 80 × 80 image, our method could run at 250 frames per second. The whole training took about 9 h and the testing about a minute. With an overall accuracy and a mean F-score of 97.2%, the proposed multi-task architecture significantly outperformed the benchmarked MLP and R-CNN models. We report in Table 4 the confusion matrix and additional accuracy metrics. Interestingly, the classification performances were relatively homogeneous across ship types (mean accuracy above 92% for all classes). Tankers involved a greater misclassification rate with some confusion with cargo.

**Table 3.** Confusion matrix and accuracy metrics for the R-CNN with 4 classes.

| | Confusion Matrix | | | | |
|---|---|---|---|---|---|
| Prediction <br> Ground Truth | Tanker | Cargo | Fishing | Passenger | Precision |
| Tanker | 845 | 97 | 3 | 33 | 86.40 |
| Cargo | 98 | 787 | 24 | 77 | 79.82 |
| Fishing | 2 | 9 | 891 | 51 | 93.49 |
| Passenger | 8 | 15 | 1 | 961 | 97.56 |
| Recall | 88.67 | 86.67 | 96.95 | 85.65 | |
| | Accuracy metrics | | | | |
| Label | Tanker | Cargo | Fishing | Passenger | Overall |
| IoU | 77.81 | 71.09 | 90.83 | 83.86 | 80.90 |
| F-Score | 87.52 | 83.10 | 95.19 | 91.22 | 89.26 |
| Accuracy | 93.82 | 91.80 | 97.69 | 95.26 | 89.29 |
| $\kappa$ | 0.83 | 0.78 | 0.94 | 0.88 | 0.88 |

**Table 4.** Confusion matrix and accuracy metrics for the proposed network with 4 classes.

| | Confusion Matrix | | | | |
|---|---|---|---|---|---|
| Prediction <br> Ground Truth | Tanker | Cargo | Fishing | Passenger | Precision (%) |
| Tanker | 985 | 11 | 0 | 4 | 98.5 |
| Cargo | 65 | 907 | 12 | 16 | 90.7 |
| Fishing | 0 | 2 | 998 | 0 | 99.8 |
| Passenger | 0 | 0 | 0 | 1000 | 100.0 |
| Recall(%) | 93.81 | 98.59 | 98.81 | 98.04 | |
| | Accuracy metrics | | | | |
| Label | Tanker | Cargo | Fishing | Passenger | Overall |
| IoU(%) | 92.49 | 89.54 | 98.62 | 98.04 | 94.67 |
| F-Score(%) | 96.10 | 94.48 | 99.30 | 99.01 | 97.22 |
| Accuracy(%) | 98.00 | 97.35 | 99.65 | 99.50 | 97.25 |
| $\kappa$ | 0.95 | 0.93 | 0.99 | 0.99 | 0.97 |

Regarding length estimation performance, our framework achieved very promising results. The length was slightly over-estimated (mean error: 4.65 m $\pm$ 8.55 m), which is very good regarding the spatial resolution of the Sentinel-1 SAR data (10 m/pixel). To our knowledge, this was the first demonstration that reasonably accurate ship length estimates could be derived from SAR images using learning based schemes, whereas previous attempts using model driven approaches led to much poorer performance. Overall, the results of the classification and length estimation tasks for all the tested architectures are summarized in Table 5.

We also trained our model with five classes, and it confirmed that our framework performed well. The length was slightly over-estimated (mean error: 1.93 m $\pm$ 8.8 m), and the classification was also very good (see Table 6). Here, we still report some light confusion for the tanker and cargo classes. The accuracy metrics were slightly worse than the four class model, but we still report an overall accuracy and a mean F-score of 97.4%.

**Table 5.** Results of all the tested architectures for the classification (4 classes) and length estimation.

| Architecture | Length Mean Error (m) | Classification Overall Accuracy (%) |
|---|---|---|
| MLP | $-7.50 \pm 128$ | 25.00 |
| R-CNN | - | 88.57 |
| Our network | $4.65 \pm 8.55$ | 97.25 |

**Table 6.** Confusion matrix and accuracy metrics for the proposed network with 5 classes.

| | Confusion Matrix | | | | | |
|---|---|---|---|---|---|---|
| **Ground Truth**    Prediction | Tanker | Cargo | Fishing | Passenger | Tug | Precision (%) |
| Tanker | 771 | 28 | 0 | 1 | 0 | 96.38 |
| Cargo | 60 | 732 | 3 | 3 | 2 | 91.50 |
| Fishing | 0 | 1 | 799 | 0 | 0 | 99.88 |
| Passenger | 3 | 1 | 0 | 796 | 0 | 99.50 |
| Tug | 0 | 0 | 0 | 0 | 800.0 | 100.00 |
| Recall(%) | 92.45 | 96.06 | 99.63 | 99.50 | 99.75 | |
| Accuracy metrics | | | | | | |
| Label | Tanker | Cargo | Fishing | Passenger | Tug | Overall |
| IoU(%) | 89.34 | 88.19 | 99.50 | 99.00 | 99.75 | 95.16 |
| F-Score(%) | 94.37 | 93.73 | 99.75 | 99.50 | 99.88 | 97.44 |
| Accuracy(%) | 97.70 | 97.55 | 99.90 | 99.80 | 99.95 | 97.45 |
| $\kappa$ | 0.93 | 0.92 | 1.00 | 0.99 | 1.00 | 0.97 |

We further analyze the proposed scheme and the relevance of the multi-task setting, compared with task specific architectures. To this end, we performed an ablation study and trained the proposed architecture using (i) length estimation loss only, (ii) classification loss only, and (iii) the combination of length estimation and classification losses (i.e., without the detection loss). We report in Table 7 the resulting performances compared to those of the proposed end-to-end learning strategy. Regarding the classification issue, combined losses resulted in an improvement of about 1.3% (above 25% in terms of relative gain). The improvement was even more significant for length estimation with a relative gain in the mean error of about 36%. Interestingly, we noted that the additional use of the detection loss also greatly contributed to the improvement of length estimation performance (mean error 2.85 m without using the detection loss during training vs. 1.93 m when using jointly detection, classification, and length estimation losses). As an illustration of the detection component of the proposed architecture, we illustrate in Figure 7a the detection result. As mentioned above, the thorough evaluation of this detection model was not the main objective of this study. Furthermore, without any precise ship footprint ground truth, it was impossible to evaluate quantitatively the performance of the network for this specific task. Let us recall that the detection task has been widely addressed in the literature [14–16]. Overall, this complementary evaluation supported the idea that neural network architectures for SAR image analysis may share some low level task independent layers, whose training can highly benefit from the existence of multi-task datasets.

**Table 7.** Ablation study; performance of the network for different scenarios: (i) only length estimation, (ii) only classification, and (iii) length estimation and classification without detection.

|  | Length Mean Error (m) | Classification Overall Accuracy (%) |
|---|---|---|
| (i) | $3.07 \pm 9.0$ | - |
| (ii) | - | 96.10 |
| (iii) | $2.85 \pm 8.9$ | 97.50 |
| Full network | $1.93 \pm 8.8$ | 97.45 |

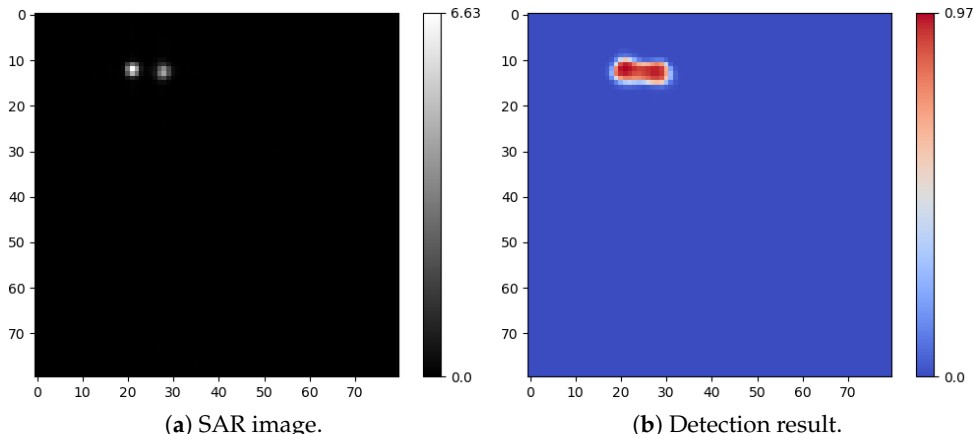

(**a**) SAR image.　　　　　　　　　　　　　　(**b**) Detection result.

**Figure 7.** Example of detection output for the considered multi-task architecture: (**left**), SAR image (with backscatter intensity) used as the input; (**right**), the output of the detection module of the considered architecture.

### 3.4. Application to a Full SAR Image

We illustrate here an application of the proposed approach to a real SAR image acquired on 4 April 2017 in Western Brittany, France. We proceeded in several steps as follows. First, a CFAR based ship detector was applied. Then, for each detected ship, we applied the trained deep network model to predict the ship category and its length. For illustration purposes, we report in Figure 8 the detected ships that could be matched to AIS signals.

For the considered SAR image, among the 98 ships detected by the CFAR based ship detector, 66 ships had their length documented, and 69 ships belonged to one of the five proposed classes after AIS matching. We may point out that the tug class is not represented. We report classification and length estimation performance in Table 8. Ship classification performance was in line with the performance reported above. Regarding length estimation, the mean error 14.56 m $\pm$ 39.98 m was larger than that reported for the ground truthed dataset. Still, this error level was satisfactory given the pixel resolution of 10 m of the SAR image. Let us note that, given the limited samples available, the standard deviation was not fully relevant here. While a special care was undertaken for the creation of our SAR-AIS dataset, this application to a single SAR image exploited the raw AIS data. AIS data may be significantly corrupted, which may partially explain these differences.

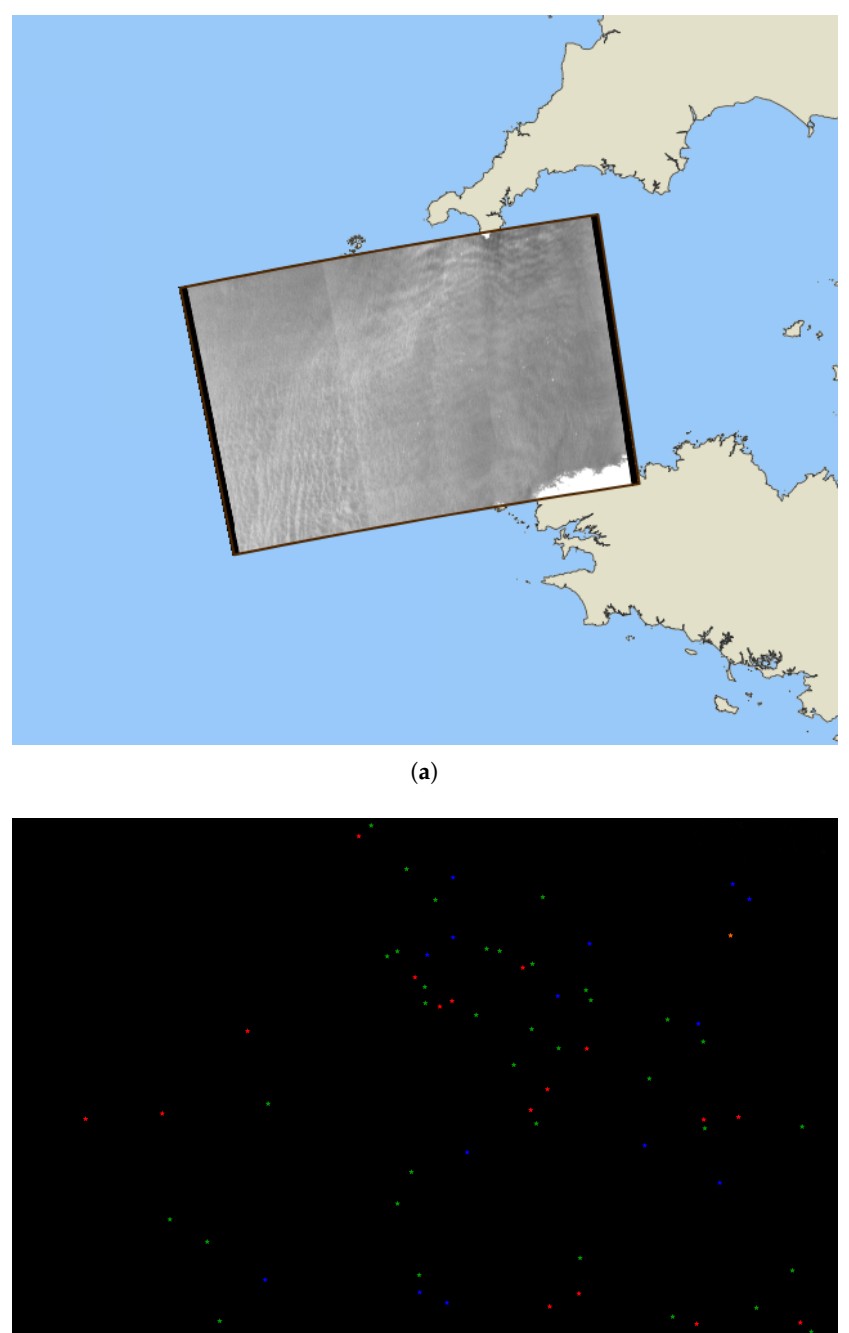

**Figure 8.** SAR image acquired on 4 April 2017 in Western Brittany, France. (**a**) Image location. (**b**) SAR image with ships identified from the matching of CFAR based detection and the Automatic Identification System (AIS). ⋆ tanker, ⋆ cargo, ⋆ fishing, ⋆ passenger (best viewed in color).

**Table 8.** Classification scores of the proposed network on small patches extracted from a SAR scene.

| Confusion Matrix | | | | | |
|---|---|---|---|---|---|
| Prediction<br>Ground Truth | Tanker | Cargo | Fishing | Passenger | Precision (%) |
| Tanker | 13 | 3 | 0 | 2 | 72.22 |
| Cargo | 5 | 30 | 0 | 0 | 85.71 |
| Fishing | 0 | 0 | 14 | 0 | 100.00 |
| Passenger | 0 | 0 | 0 | 1 | 100.00 |
| Recall | 72.22 | 90.91 | 100.00 | 33.33 | |
| Accuracy metrics | | | | | |
| Label | Tanker | Cargo | Fishing | Passenger | Overall(%) |
| IoU | 56.52 | 78.95 | 100.00 | 33.33 | 67.20 |
| F-Score | 72.22 | 88.24 | 100.00 | 50.00 | 77.61 |
| Accuracy | 85.29 | 88.24 | 100.00 | 97.06 | 85.29 |
| $\kappa$ | 0.62 | 0.76 | 1.00 | 0.49 | 0.85 |

### 3.5. Application to the OpenSARShip Dataset

The OpenSARShip dataset [29] has recently been made available to the community. We report here the results obtained with our framework when applied on this dataset. This dataset comprises SAR data with different polarization characteristics and also includes ship categories. With a view toward easing the comparison with the previous results, we focused on SAR images that were in VV polarization and the ship categories considered, which led to considering the following four categories; tanker, cargo, fishing, and passenger. Overall, we considered a dataset of 5225 ships (80% were employed for training and 20% for testing). In the OpenSARShip dataset, the classes are not equally represented. We report classification and length estimation performance in Table 9. We also evaluated the performance of the model trained on our dataset and applied on the OpenSARShip dataset and conversely.

The results showed that our model produced good results when trained and tested on the same database. However, the results did not transfer from one dataset to another. We suggest that this may relate to differences in the maritime traffic and environment between Europe (our dataset) and Asia (OpenSARShip dataset). The comparison to previous work on the OpenSARShip dataset was not straightforward. For instance, the work in [36] considered only a three class dataset (tanker, cargo, and other). The reported accuracy score (76%) was lower than our 87.7% accuracy score for the considered five class dataset. We may also emphasize that [36] did not address ship length estimation.

**Table 9.** Comparison of the results of the network on our database and on the OpenSARShip database.

| Test | | Train | | | |
|---|---|---|---|---|---|
| | | Ours | OpenSARShip | Ours | OpenSARShip |
| | Ours | 97.45 | 22.18 | Ours 1.93 ± 8.8 | 56.78 ± 314.78 |
| | Ours | 34.05 | 87.71 | Ours −102.51 ± 123.94 | −0.23 ± 11.04 |
| | Classification overall<br>accuracy (%) | | | Length mean<br>error (m) | |

## 4. Discussion

The reported results showed that a dedicated architecture was necessary for ship classification and length estimation, while state-of-the art architectures failed to achieve satisfying performances. The MLP was sufficient for ship detection on SAR images (from a visual assessment). However, this should not be

considered as a good result since we only had (positive) examples of ships in our database (no negative samples, so we could not assess the false positives). Thus, the network only learned a thresholding and could not discard a ship from other floating objects (e.g., icebergs). Indeed, iceberg detection and discrimination between iceberg and ship are specific research questions [37,38]. Overall, the performance of the MLP stressed the complexity of the classification and length estimation tasks. In terms of classification accuracy, the R-CNN performed better than the MLP, with an overall accuracy of 88.57%. These results support the proposed architecture with three task specific networks, which shared a common low level network. The latter was interpreted as a feature extraction unit that the task specific networks relied on.

Compared to the state-of-the art architectures (MLP and R-CNN), our model produced better results for ship classification and length estimation from Sentinel-1 SAR images with only a few confusions between classes. A multi-task architecture was well adapted for simultaneous ship classification and length estimation. Our model also performed well when a new class was added (e.g., tug). Furthermore, adding a detection task (even with a coarse ground truth) tended to improve the length estimation. Our experiments also showed that the trained models did not transfer well from one dataset to another. We suggest that this may relate to differences in the characteristics of the maritime traffic and/or marine environment. Future work should further explore these aspects for the application of the proposed model worldwide.

## 5. Conclusions

In this paper, a multi-task neural network approach was introduced. It jointly addressed the detection, classification, and length estimation of ships in Sentinel-1 SAR images. We exploited synergies between AIS and Sentinel-1 to build reference datasets automatically for training and evaluation purposes, with the ultimate goal of relying solely on SAR imagery to counter a lack or corruption of AIS information that corresponds to illegal activities. While the polarization type had a significant effect on SAR backscatter, we were able to train a model that jointly processed HH or VV polarization without prior information on the type of polarization. Our results supported the assumption that HH and VV polarizations shared common image features and that differences in backscatter distributions could be handled through an appropriate parameterization of the network.

Regarding the considered architecture, a mutual convolutional branch transformed raw inputs into meaningful information. Such information was fed into three task specific branches. Experimental evaluation showed improvement over standard MLP or R-CNN. Ship detection cannot be totally assessed, but a visual inspection supported the relevance of this detection stage. Besides, it was shown to significantly contribute to improved performance of the classification and length estimation components. Overall, we report the promising performance for ship classification (above 90% of correct classification) and length estimation (relative bias below 10%). Considering a residual architecture appeared as a critical feature to reach good length estimation performance, this would require further investigation.

Future work may further investigate the training and evaluation of the detection stage. The automation of the matching process between AIS data and SAR images has the potential for significantly increasing the size and diversity of the training and evaluation datasets. This may provide new avenues to address generalization and transfer issues between geographic areas pointed out in our results. Furthermore, while SAR imagery is less affected by weather conditions than optical imagery, a specific analysis of the impact of weather conditions on the identification performance would also be of interest. Finally, the specificity of the SAR imagery would call for dedicated operations, while our network relied on standard techniques issuing from computer vision.

**Author Contributions:** C.D. carried out the experiment and wrote the manuscript with support from S.L. and R.F., R.V. and G.H. provided the SAR data. All authors contributed to the final manuscript.

**Funding:** This research was funded by ANR (Agence Nationale de la Recherche) SESAME (ANR-16-ASTR-0026).

**Conflicts of Interest:** The authors declare no conflict of interest.

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
