# Peer review of "Ship Identification and Characterization in Sentinel-1 SAR Images with Multi-Task Deep Learning"

_remotesensing, doi:10.3390/rs11242997_

Round 1

Reviewer 1 Report

Please see detailed comments attached

Author Response

We thank all Guest Editors and Reviewers for their in-depth analysis of our paper as well as their fruitful suggestions. We have revised our manuscript accordingly and we provide below a point-by-point response to reviewers' comments.

Reviewer 2 Report

Summary

The manuscript proposes an effective algorithm for ship detection, classification and length estimation based on AIS and SAR Sentinel-1 data.

Broad comments

The manuscript is clearly structured, the methods proposed by the authors are described in detail, and the results are compared with the works of other authors. The study is performed and presented at a high scientific level.

Specific comments

Despite the overall high level of the study, there are the following comments.

15/16 “network provides satisfactory results, with accurate classification … ”: Please be precise and provide a quantitative measure of the result quality.

35 “[3,4] extracted handcrafted features” and further in the text: if you use reference at the beginning or in the middle of the sentence please write down the name of the author followed by the reference, for example: "Alshammari et al. [25] detected subsidence...".

46 “it is not critically affected by weather conditions”: Were the wind speed and wave height estimated? And how they may affect the ships detection and classification quality?

81/82 “mean absolute error: 30m +/- 36.6m”: It would be interesting to know the range of actual ship lengths. What percentage of the actual length is the mean absolute error?

116/117 “a sigmoid for the detection, a softmax activation for the classification, and a linear activation”: Please explain the choice of activation functions.

130 “(see Figure ??)”: Please provide the Figure number.

138 “SAR images”: Did you use raw data or the SAR images were preprocessed?

174 “The MLP is however sufficient for ship detection”: Please provide the overall accuracy for ship detection.

203 “Coefficient”: Please use lower case.

215 Figure 6: Please provide the reference before the Figure appear in the text.

261 Figure 7: Please represent also the study area - a basemap or a SAR image, for example:
Figure 2 of this paper: https://www.mdpi.com/2072-4292/11/18/2171
Figure 11 of this paper: https://www.mdpi.com/2072-4292/11/7/765
Include a scale bar, a north arrow, and coordinates. Provide Sentinel-1 image acquisition dates. It is difficult to distinguish Tanker and Passenger - red and orange colors look similar. Ship markers are too small, which complicates the Figure perception.

261 Table 5: Please explain the “P0” and “Pe” metrics.

Author Response

(The authors gave the same response as above.)

Reviewer 3 Report

This work is important for ship ID and classification in SAR images using deep learning. It includes training on a vast database of ships and Sentinel-1 satellite images. Extensive work.

The analysis could be improved in several ways:

The four ship classes are fine but what about the more abundant smaller boats and sailing ships? They are also interesting because they test the limit of your resolution and thus your detection and classifications algorithms.  Your loss function has three quite different parts. Why are they simply added? One might give them different weighs in the total loss and optimize their weight thereby improving the resulting confusion matrix. How is the class loss correlated with the length loss? One might expect that the four classes have different average length? It would be nice with a figure of all ships in a L_true vs L_class in a scatter plot. The four classes in four colors. This might answer the question above. Line 130 has a Figure??? reference. Line 140: the HH and VV can have quite different backgrounds and ship reflections. Deep learning algorithms are known to fail when training on different databases. How does this influence the results? Line 172: by "important" - do you mean "large" ? Line 173: by mean error, do you mean that -7.5m is the mean difference between detected and true length, and that +/-128m is the standard deviation? What is understood by the authors may not be well known to the general reader. The mean difference between detected and true lengths vary between various deep learning algorithms. The difference is of order the spatial pixel resolution. When the deep learning algorithm has thousands of parameters adjusted to minimize the differences, I would expect that the difference should be exactly zero to order 1/N, where N is the total number of ships? The confusion matrix in table 1 is almost diagonal, except for the 65 misclassified tanker/cargo ships. Why? Are their size the same or their reflection? Black ships (no AIS signal) should be discussed. Are they excluded? This question is related to the small ships mentioned above, which do not have to use AIS transponders. These questions are therefore also related to ship size. Line 247: the authors honestly report a large mean error. Their excuses (pixel resolution) are not large for all other lengths, and the few ships (66) should only result in a much smaller statistical error. How do the CFAR depend on satellite viewing (inclination) angle? The background changes significantly which affects deep learning algorithms.

Author Response

(The authors gave the same response as above.)

Round 2

Reviewer 3 Report

No further